# Sociodemographic Characteristics of Traditional Healers and Their Knowledge of Noma: A Descriptive Survey in Three Regions of Mali

**DOI:** 10.3390/ijerph16224587

**Published:** 2019-11-19

**Authors:** Denise Baratti-Mayer, Moussa Baba Daou, Angèle Gayet-Ageron, Emilien Jeannot, Brigitte Pittet-Cuénod

**Affiliations:** 1Service of Plastic, Reconstructive and Aesthetic Surgery, Geneva University Hospitals, 1205 Geneva, Switzerland; 2Geneva Study Group on Noma (GESNOMA), University of Geneva, 1205 Geneva, Switzerland; 3Centre Hirzel, 91093 Bamako, Mali; daoumoussa07@gmail.com; 4Service of Clinical Epidemiology, Geneva University Hospitals, 1205 Geneva, Switzerland; angele.gayet-ageron@hcuge.ch; 5Institute of Global Health, Faculty of Medicine, University of Geneva, 1202 Geneva, Switzerland; emilien.jeannot@unige.ch; 6Addiction Medicine, Department of Psychiatry, Lausanne University Hospital and University of Lausanne, 1004 Lausanne, Switzerland

**Keywords:** access to health, cancrum oris, Noma, traditional healers, traditional medicine, west Africa

## Abstract

*Background*: Noma can be a lethal disease and causes disfigurement in young children in low-resource countries, particularly in Africa. In these settings, 80% of the population mainly consult traditional healers for healthcare problems. Our study aimed to describe the sociodemographic characteristics of traditional healers and to assess their knowledge of noma. *Methods*: We conducted a survey among traditional healers in three Malian regions from May 2015 to January 2016 and collected data on sociodemographic characteristics, professional activity, knowledge, and experience of noma and collaboration with modern medicine. *Results*: Among 770 traditional healers invited to participate, 732 responded (95%) (mean age, 54.5 years). Most were illiterate (66.3%), which was associated with older age (*p* < 0.001). Although they treated all types of disease, only 10.5% had some knowledge of noma, with regional differences (*p* < 0.001). *Conclusion*: Noma is poorly known among traditional healers, especially in remote areas. Our findings suggest a lack of interest among young people for traditional medicine, implying an imminent decrease of healers, and thus the need for national health systems to strengthen and promote access to modern health care. Training programmes to improve the early diagnosis referral of noma patients should include all types of primary health workers.

## 1. Introduction

Noma is a fulminating gangrenous lesion causing death or severe disfigurement in young children living in the poorest and most remote areas of developing countries, with lethality estimated to be as high as 80% [1,2]. Survivors suffer not only from aesthetic prejudice, but also severe functional impairment related to the destroyed facial structures, as well as social stigma and discrimination due to local popular beliefs that attribute noma to evil spirits [3]. A bacterial aetiology of noma has been long investigated. However, recent studies using state-of-the art microbiological techniques were unable to identify a strictly noma-linked bacteria and the results suggested that an imbalance in the oral microflora appeared to be associated with the presence of noma disease [4,5].

Our previous studies have shown that chronic malnutrition, a recent infectious disease, immunodeficiency, and a lack of oral hygiene are risk factors for noma [6,7]. Noma is preceded by a particular form of gingivitis, acute necrotizing gingivitis (ANG), which is characterized by spontaneous gingival bleeding, necrosis, and pain [7,8]. In developed countries, ANG is exceptional in children, affecting mainly young adults and associated with immunodeficiency, substance abuse, stress, and bad oral hygiene [9]. By contrast, ANG can affect up to 50% of children <12 years in Africa, depending on the region. In these settings, ANG seems to be linked to inadequate oral hygiene and severe poverty [10,11]. Unlike simple gingivitis, which is easily manageable by improving oral hygiene, ANG should be treated by professional oral hygiene methods, sometimes requiring antibiotics. If left untreated, it can evolve towards noma, especially when the child is in poor health. For this reason, the World Health Organization (WHO) has recently defined ANG as the first stage of noma (stage 1). The disease then evolves through different phases: Oedema (stage 2); necrosis (stage 3); scarring (stage 4); and sequelae (stage 5) [12] (Figure 1). The pathogenesis of the progression from ANG to noma remains unclear. Some bacteria have been described as trigger organisms, but these have not been further confirmed [5,13].

Treatment of acute noma consists of intravenous hydration, nutrition, and antibiotics to avoid lethal complications, such as sepsis and pneumonia, and attempts to contain the disease [14]. When applied at stages 1 or 2, this treatment can interrupt the dramatic evolution and avoid facial mutilation and consequent functional impairment and social stigma. Later stages result in irreversible tissue destruction for which the only answer is complex reconstructive surgery that is not always feasible in the countries where the children live [14].

Our previous epidemiological and microbiological research convinced us that preventive and information actions were needed to reduce the lethality and morbidity of noma in affected countries. To this end, an early diagnosis represents the “foundation stone” of this strategy. Noma has already been recognized by WHO as a public health problem in 1994. Since 2003, six countries in West Africa (including Mali) have launched national programmes against noma, with the aim to increase knowledge of the disease among primary “modern” healthcare workers and thus increase the early detection rate and early treatment [15].

However, according to WHO, 80% of the rural population in low-income countries will first consult a traditional healer for any problem related to primary health care [16]. Traditional medicine is the oldest known healthcare system and was the only answer to health problems in Africa before colonization. It is still used by millions of people around the world. Traditional medicine tends to consider a disease as the rupture of an equilibrium, i.e., a harmony between body and soul [17]. The preferential use of traditional medicine can be explained by the lack of any other choice (difficult access to modern health care for geographic or financial reasons) and by cultural beliefs [18]. As shown by Abdullahi, the ratio of traditional healers to the population in Africa is 1:500 compared to 1:40,000 for “modern healthcare workers” [17]. A previous study conducted in Niger in a region benefiting from awareness-raising programmes showed that 40% of children affected by noma have been exposed to traditional medicine before seeking “modern” medical advice [6]. This proportion may be even higher in regions where no information campaigns have been conducted. Therefore, it appeared useful to explore the actual extent of knowledge of the disease among traditional healers, who are often the first to be consulted.

## 2. Objectives

The aims of our study were to describe the sociodemographic characteristics of traditional healers in three regions of Mali in order to understand their background and way of working, with a particular focus on their knowledge of noma and implication in its diagnosis and treatment. We also aimed to evaluate their capacity/willingness to participate in teaching programmes and to collaborate with modern medicine in order to improve disease management at the primary care level.

## 3. Methods

### 3.1. Study Design, Settings, and Participants

We conducted a descriptive cross-sectional study from May 2015 to January 2016 (9 months), among traditional healers living and working in three distinct regions (departments) in Mali, i.e., Kayes, Koulikoro, and Sikasso (Figure 2). As no formal data were available regarding the prevalence of noma in Mali, we were unable to include this as a criterion and the three regions were selected because they are representative of Mali in terms of climate, agriculture, medical facilities, and population, but also for accessibility and safety reasons as access to some regions has been prohibited due to sociopolitical insecurity.

Traditional healers may have different specialties, such as herbalist (who knows the therapeutic properties of herbs and sells them), healer (who combines inherited traditional knowledge and herbs), bonesetter or spiritualist (who treats using “supernatural” or religious power). Traditional healers are organized in a national federation in Mali since 2002, which provided us with their membership list. We invited those living and working permanently in one of the three regions to participate. The Ministry of Health and the National Federation of Traditional Healers and Herbalists of Mali (“Fédération Nationale des Tradipraticiens et Herboristes du Mali”) approved the study protocol and provided the authorization to conduct the survey.

### 3.2. Data Collection

A dedicated questionnaire was designed for the study and comprised 25 questions divided into five sections to investigate their sociodemographic characteristics, professional activity, knowledge and professional experience of noma and gingivitis, and the extent of their collaboration with modern medicine. Fifteen questions had predefined answers and the remaining 10 questions were open. The questionnaire also attempted to evaluate the potential interest of traditional healers in teaching programmes on noma. It was read by the field investigator (MBD) and translated when necessary. For each region, different meetings of approximately 20–25 participants were organized by the president of the national federation, together with the president of each local association. The days and place of the meetings were communicated to participants by the local president by telephone and/or public radio. During each meeting, the investigators first gave explanations regarding the study’s aim and modalities. To ensure that questions would be correctly understood and that the definition of the disease was the same for all participants, healers were shown pictures of the different stages of the disease prior to completion of the questionnaire. After this introduction, healers were interviewed individually by investigators in order to complete the questionnaire (approximately 30 min per questionnaire). Following completion, participants received a reimbursement of travel expenses and a per diem.

### 3.3. Data Analysis

We used a convenience sample of traditional healers affiliated to the federation and who accepted to participate in our survey. Categorical variables were presented by their frequencies and relative percentages. Continuous variables were presented by their mean, standard deviation, and median. We compared sociodemographics, professional activities, and medical knowledge between the three regions. Categorical variables were compared using the Chi square test or Fisher’s exact test, depending on the application criteria. Continuous variables were compared using ANOVA or the Kruskal–Wallis nonparametric test. We also compared the educational level (no education, primary, secondary, superior, and other) by categories of age (≤ and >30 years) using the Chi square test. A *p* value < 0.05 was considered statistically significant. Statistical analyses were performed with Stata 14 IC (StataCorp. 2015, Stata Statistical Software: Release 14, College Station, TX, USA).

## 4. Results

Among 770 traditional healers invited, 732 (95%) accepted to participate: 197 in Kayes (26.9%); 209 in Koulikoro (28.6%); and 326 in Sikasso (44.5%). Sociodemographic characteristics of traditional healers (overall and by region) are presented in Table 1. Participants had a mean age of 54.5 years and the majority were men, Malians and Muslims. Most traditional healers were illiterate. Only 10.5% had completed mandatory education (primary and secondary levels, up to 15 years) and 5.6% had reached a superior level of education (Table 1). Age was associated with the education level (*p* < 0.001), which was higher among younger healers. Among those ≤30 years (*n* = 39), 46.2% had completed mandatory education (primary and secondary level) and 23.1% had a superior level compared with 27.0% and 4.5%, respectively, among those >30 years (*n* = 662). Indeed, among healers >30 years, 68.4% declared to have received no education, compared to 30.8% for those ≤30 years (data not shown). Regarding the professional activity, most (74.8%) practiced traditional medicine by combining the two specialties “herbalist and healer”; 50.9% had agriculture as a second professional activity. Most traditional healers had acquired training by heredity (familial tradition) or through teaching by a “master”. Types of traditional practices and secondary professional activities varied between the three regions (*p* < 0.001).

The three most frequent diseases seen by traditional healers were digestive problems, malaria, and neuropsychological problems (51.9% of all diseases declared). Oro-dental problems concerned 4.2% of indications treated (Table 2).

We did not observe any difference in the treatments dispensed to patients, but most (98.2%) were the prescription of herbs (leaves, plant extracts, roots, etc.) (Table 3).

The reasons given for collaborating with modern medicine were for treatment (90.5%), clinical diagnosis (29.4%) and follow-up (18.6%), and these differed significantly between the three regions (Table 4).

Only a few participants had some knowledge of noma disease (10.5%) or other gingival diseases (9.6%). Differences were also observed between regions: In Koulikoro, 28.9% (*n* = 59) and 25.5% (*n* = 53) of healers had knowledge of noma and gingival diseases, respectively, compared to only 4.0% (*n* = 13) and 4.3% (*n* = 14), respectively, in Sikasso, and 1.5% (*n* = 3) for both in Kayes (*p* < 0.001). Based on healers’ answers, the number of noma and ANG cases followed were very low. Interestingly, we observed the same regional differences with significantly more cases seen by healers from Koulikoro compared to Kayes and Sikasso (Figure 3). None of the healers reported having followed patients with noma sequelae. Treatments reported were mainly based on herbal and plant therapies. Regarding a potential interest in receiving some education specifically on noma and gingivitis, all (100%) were in agreement, including a willingness to have some collaboration with modern medicine in the case of a suspicion of noma among their patients.

## 5. Discussion

To our knowledge, this is the first study reporting sociodemographic characteristics of traditional healers in Mali and evaluating their knowledge on noma and gingivitis. Since 1978, following the commitment of WHO to promote the integration of traditional healers in national programmes [19] actions have led to the creation in 1986 in Mali of a Department of Traditional Medicine attached to the National Institute of Research in Public Health. This was followed in 2002 by the creation of regional federations. Of note, our study would have been impossible without the help of the national federation and the local associations, which facilitated access to traditional healers.

Our sociodemographic survey showed a high proportion of illiteracy among most traditional healers. This is directly related to the mean age of our study population, which is quite old when considering current life expectancy in Mali (67 years in 2013) [20]. Although the literacy rate and level of education were particularly low, it was significantly higher for younger (<30 years) traditional healers. Of note, the literacy rate in Mali has continuously increased from 9.5% in 1976 for individuals aged >15 years to 33.1% in 2015 [21]. This reflects the progressive evolution of the literacy rate at primary education level in Mali since the 1960s and the associated increase at secondary education level since the early 1990s [22]. In addition, the relatively old mean age of participants (54.5 years) may suggest a lack of interest from younger generations for this professional activity as recently reported in Ghana [23]. These findings allow to hypothesize that future healers could be fewer in number, but with a higher education level and therefore more prone to collaborate with modern medicine. On the other hand, the lack interest of traditional medicine among young people may imply an imminent decrease of healers and thus the need to strengthen and promote access to modern health care.

Our results show some differences between regions regarding specialty, training, and second professional activity. When traditional healers have a second professional activity that is mainly agriculture, they often combine “inherited knowledge” with “herbalism”. This knowledge of healing herbs that they cultivate and sell is taught by parents or relatives (heredity/family tradition). In less agricultural and more remote regions, most traditional practitioners have no other professional activity and are “only” healers using an inherited or “supernatural” power sometimes acquired by a personal teacher.

Our survey showed that traditional healers are involved in the care of all types of disease, both psychological and somatic, including birthing, bone fractures, or dental problems.

We observed that a group of four diseases (malaria, digestive, neuropsychological, and genital problems) represented 62% of consultations. Among the remaining cases, oro-dental problems were the fourth most frequently reported cause of consultation. The reporting of consultations with healers for classic childhood diseases or for trauma were surprisingly low. Thus, it can be hypothesized that it has now been integrated that modern medicine proposes better care management in health facilities for such conditions, even in remote areas. Traditional healers use herbalism and prescribe herbs and plant extracts as treatment. Although these treatments are useful for a large number of diseases and especially for malaria, [24] this is not the case for acute noma, which needs prompt referral for life support and antibiotics.

Reports on African traditional healers are quite rare and focus essentially on their knowledge and treatment of psychiatric disorders or frequent endemic diseases, such as malaria, tuberculosis, or human immunodeficiency virus [25,26,27,28,29]. To date, and to our knowledge, there are no studies on traditional healers and noma disease. Our survey showed that most traditional healers had a poor knowledge of noma. Regional differences can be explained by some geographical aspects. For example, Koulikoro is close to Bamako where most of the nongovernmental organizations dealing with noma are located and where national information programmes and prevention actions are conducted. Thus, this may explain the higher proportion of healers in this area with some knowledge of noma compared to the two other regions. Traditional healers with some knowledge of noma reported having been consulted more often for noma cases than those without any previous awareness of the disease. In our opinion, the higher prevalence of noma in Koulikoro compared to Kayes and Sikasso, is only apparent and is probably linked to a better knowledge and therefore a better recognition of the disease. The overall number of cases declared to have been seen by traditional healers was very low.

This may first reflect misdiagnosis. Noma is often confused with other conditions, especially with tumours or dental abscesses in its oedema stage, and with cleft lip or other congenital craniofacial malformations in the sequelae stage, therefore potentially leading to an underestimation of cases. The low incidence of acute noma cases reported by traditional healers could also be related to the fact that parents seem to favour “modern” medicine in the case of diseases affecting young children. This finding highlights the need for information and teaching programmes on noma across all levels of primary healthcare workers. This is confirmed by a recent publication reporting poor practice competences in noma care among “modern” primary healthcare workers in Burkina Faso [30]. Our results can also be compared to those reported in a study in Cameroon investigating the role of traditional healers in the diagnosis and treatment of Burkitt lymphoma, a cancerous condition affecting children in West Africa. Similar to the knowledge of noma among Malian traditional healers, those in Cameroon were ignorant about Burkitt lymphoma and contributed to a late diagnosis of the disease [31].

### Limitations

Our study has some limitations. First of all, the social desirability bias inherent to any survey design and the inclusion criteria (i.e., having accepted to participate and belonging to a professional federation) implies a convenience sample and a preselection of the study population. It is known that many traditional healers work in an independent and nomadic way, without recognition by the national federation. Therefore, we cannot exclude that their sociodemographic characteristics, knowledge of noma, or the number of noma cases seen by them could be different and our sample may not be representative of all traditional healers in Mali. Second, since the study design included a reimbursement of expenses and a per diem for participants, we cannot exclude that this financial aspect constituted an inclusion bias, which could also explain the high participation rate and high interest in participating in future training. Third, language may have represented another limitation as linked to the necessity of using a translator. Fourth, we cannot exclude the existence of some information bias, in particular recall bias and the lack of statistics compiled by the healers about their daily activity. Finally, problems related to an existing rivalry and distrust between traditional and modern medicine could also have represented a limitation. Despite the initial explanations about the study objectives, some traditional healers were suspicious about the investigators’ intentions and we cannot exclude that some answers may have been intentionally wrong. Irrespective of these limitations, this work is the first sociodemographic description of traditional healers in Mali and confirms their implication at the primary care level, as well as highlighting their considerable lack of knowledge regarding noma.

Given the role of traditional healers, it seems essential to integrate them in the process of medical management at the primary health care level. Since the commitment of WHO to promote their integration in national programmes [19] there has been some collaboration between the two medicines and an increasing number of national federations of traditional healers recognized by the ministries of health are emerging in different countries in West Africa [18,23]. Recently, the development of teaching programmes for traditional healers has been increasingly supported and recent reports on their implication in human immunodeficiency virus prevention/care are promising [29,32]. The development of teaching programmes for healers is difficult and represents a considerable organizational and financial effort in countries with limited resources and many other competing healthcare priorities. Nevertheless, with the help of the national federations, it should be possible to identify younger traditional healers who could become interlocutors and “teachers” within the context of these programmes. Regarding noma disease, which mainly affects very young children, teaching programmes should be directed towards modern healthcare workers, traditional healers, as well as midwives and matrons.

## 6. Conclusions

There appears to be a lack of attractiveness for traditional medicine among young generations in Mali and this needs to be further investigated. If confirmed, a decrease in healers can be expected in the near future, which implies a pressing need to broaden the range of modern medical care offered at present in order to respond to the population’s needs. National health systems should be prepared to bridge this potential gap and to ensure adequate care not only for noma, but also for other diseases.

## Figures and Tables

**Figure 1 ijerph-16-04587-f001:**
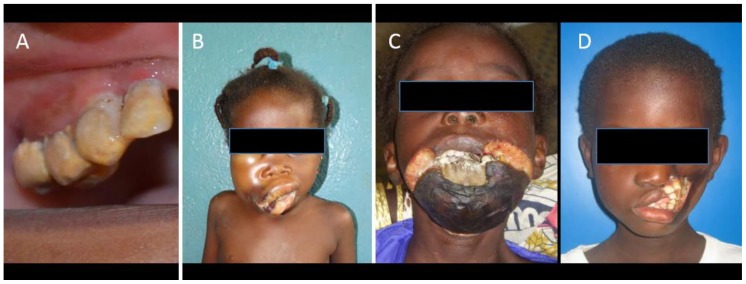
(**A**) Acute necrotizing gingivitis with the presence of a huge amount of calculus, greyish deposits, and decapitation of gingival papillae; (**B**) stage 2 (oedema) in a young Malian girl. Oedema of the right cheek can be observed with incipient necrosis starting at the lip angle; (**C**) stage 3 (necrosis) in a Malian boy. The chin and the adjacent region are necrotic, with exposure of mandibular necrotic bone in the middle. The detachment of the margins of the necrotic tissue from the surrounding healthy regions can be observed; (**D**) stage 5 (sequelae) in a girl presenting loss of tissue affecting the left cheek and both lips.

**Figure 2 ijerph-16-04587-f002:**
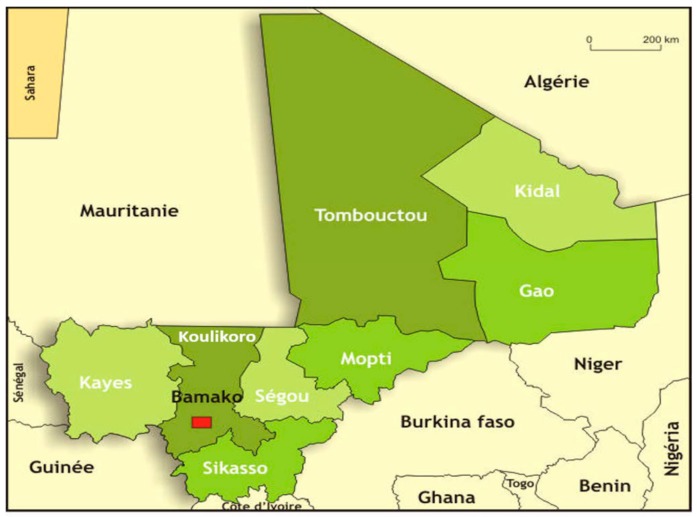
Geographical map of Mali showing the three study regions: Kayes, Koulikoro, and Sikasso, and the capital city, Bamako (located in the Koulikoro region). (Image modified from “Bamada.net” 10.05.2016).

**Figure 3 ijerph-16-04587-f003:**
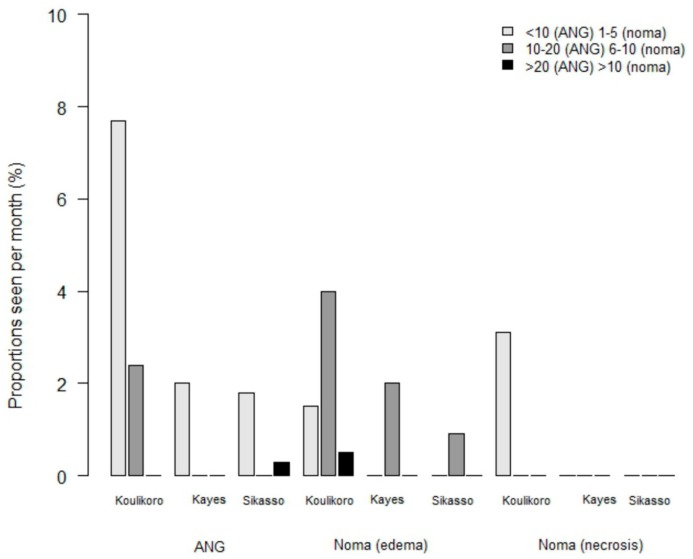
Professional experience of traditional healers in the care of noma cases. The number of monthly reported cases differs for acute necrotizing gingivitis and noma.

**Table 1 ijerph-16-04587-t001:** Sociodemographic variables.

Descriptive Variables *n* = 732	Total	Kayes *n* = 197	Koulikoro *n* = 209	Sikasso *n* = 326	*p*-Value
Men, *n* (%) (*n* = 714)	537 (75.2)	137 (69.9)	166 (81.4)	234 (74.5)	n.s.
Mean age (± SD, median) (*n* = 730)	54.5 (±14.0, 56.0)	54.6 (±13.9, 56)	57.6 (±13.1, 59)	52.5 (±14.3, 53)	<0.001
Religion, *n* (%) (*n* = 717)					<0.001
Christian	12 (1.7)	1 (0.5)	8 (4.1)	3 (0.9)	
Muslim	695 (96.9)	195 (99.5)	185 (94.9)	115 (96.6)	
Animist	8 (1.1)	0 (0)	0 (0)	8 (2.5)	
Others	2 (0.3)	0 (0)	2 (1.0)	0 (0)	
Malian nationality, *n* (%) (*n* = 719)	719 (100.0)				
* Education, *n* (%) (*n* = 703)					n.s.
No education	466 (66.3)	132 (70.2)	129 (65.5)	205 (84.5)	
Primary	123 (17.5)	31 (16.5)	34 (17.3)	58 (18.2)	
Secondary	74 (10.5)	17 (9.0)	22 (11.2)	35 (11.0)	
Superior	39 (5.5)	8 (4.3)	11 (5.6)	20 (6.3)	
Other	1 (0.1)	0 (0)	1 (0.5)	0 (0)	
Specialty, *n* (%) (*n* = 732)					
Herbalist	13 (1.8)	1 (0.5)	7 (3.3)	5 (1.5)	
Healer	111(15.2)	0 (0)	16 (7.7)	95 (29.1)	
Bonesetter	1 (0.1)	1 (0.5)	0 (0)	0 (0)	
Spiritualist	1 (0.1)	0 (0)	0 (0)	1 (0.3)	
Other	3 (0.4)	0 (0)	1 (0.5)	2 (0.6)	
Herbalist and healer	547 (74.8)	166 (84.3)	162 (77.5)	219 (67.2)	
Herbalist and spiritualist	1 (0.1)	0 (0)	1 (0.5)	0 (0)	
Herbalist, healer and bonesetter	12 (1.6)	11 (5.6)	0 (0)	1 (0.3)	
Herbalist, healer and spiritualist	33 (4.5)	16 (8.1)	14 (6.7)	3 (0.9)	
Other combinations	10 (1.4)	2 (1.0)	8 (3.8)	0 (0)	
Other profession, *n* (%) (*n* =7 32)					
Agriculture	282 (50.9)	78 (39.6)	98 (46.9)	106 (32.5)	
Housekeeping	58 (7.9)	20 (10.1)	17 (8.1)	21 (6.4)	
Merchant	47 (6.4)	18 (9.1)	11 (5.3)	18 (5.5)	
Hunter	20 (2.7)	6 (3.0)	8 (3.8)	6 (1.9)	
No other profession	178 (24.3)	32 (16.2)	33 (15.8)	113 (34.7)	
Other	147 (20.1)	43 (21.8)	42 (20.1)	62 (19.0)	
**Training in traditional medicine, *n* (%) (*n* = 732)**					
**Personal master/teacher**	249 (34.0)	44 (22.3)	46 (22.0)	159 (48.8)	
**Mystical choice**	14 (1.9)	8 (4.1)	2 (1.0)	4 (1.2)	
**Revealed**	8 (1.1)	2 (1.0)	4 (1.9)	2 (1.6)	
**Heredity**	340 (46.5)	111 (56.4)	103 (49.3)	126 (38.7)	
**Other**	5 (0.7)	1 (0.5)	2 (1.0)	2 (0.6)	
**Combination of the above**	116 (15.8)	31 (16.2)	52 (25.0)	33 (10.1)	

SD: Standard Deviation; n. s.: Not significant; * Level of education: Primary = 7–12 years; secondary = 13–15 years; superior = from 16 years—Mandatory education is up to 15 years.

**Table 2 ijerph-16-04587-t002:** Most frequent pathologies treated by the traditional healers (by category) *.

Most Frequent Pathologies: Up to three Pathologies Mentioned Per Traditional Healer, *n* (%)	Total *n* of Traditional Healers *n* = 732	Kayes *n* = 197	Koulikoro *n* = 209	Sikasso *n* = 326	*p*-Value
Digestive problems	415 (21.8)	99 (18.8)	105 (20.4)	211 (24.6)	<0.001 ^**^
Malaria	324 (17.0)	72 (13.7)	69 (13.4)	183 (21.4)	
Neuropsychological problems	247 (13.0)	88 (16.7)	67 (13.0)	92 (10.7)	
Genital problems (including births, sexually transmitted diseases, and fertility problems)	196 (10.3)	72 (13.7)	65 (10.7)	59 (8.0)	
Skin problems	180 (9.5)	57 (10.8)	77 (14.9)	46 (5.4)	
Unspecific pain and “unclear” diseases	102 (5.4)	35 (6.6)	28 (5.4)	39 (4.6)	
Infections (unless malaria and childhood diseases)	97 (5.1)	16 (3.0)	23 (4.5)	58 (6.8)	
Oro-dental problems and noma	80 (4.2)	17 (3.2)	21 (4.1)	42 (4.9)	
Respiratory problems	48 (2.5)	9 (1.7)	22 (4.3)	17 (2.0)	
Cardiovascular problems	43 (2.3)	12 (2.3)	11 (2.1)	20 (2.3)	
Eyes and ear-nose-throat	42 (2.2)	9 (1.7)	13 (2.5)	20 (2.3)	
Orthopaedics (non-traumatic)	41 (2.2)	12 (2.3)	9 (1.7)	20 (2.3)	
Urology	28 (1.5)	5 (1.0)	12 (2.3)	11 (1.3)	
Childhood diseases	28 (1.5)	10 (1.9)	4 (0.8)	14 (1.6)	
Trauma (including bone fractures)	18 (1.0)	8 (1.5)	0	10 (1.2)	
Snakebites	11 (0.6)	6 (1.1)	0	5 (0.6)	
Total	1900 (100)	527 (100)	516 (100)	857 (100)	

* Results refer to the number of times a pathology was cited as one of the three most frequent. Traditional healers could mention up to three different pathologies, therefore the total at the foot of the table refers to the number of answers. ^**^ Global *p*-value comparing most frequent pathologies among regions.

**Table 3 ijerph-16-04587-t003:** Treatments most cited by the traditional healer (for all pathologies) *.

Descriptive Variables	Total *n* = 732	Kayes *n* = 197	Koulikoro *n* = 209	Sikasso *n* = 326	*p*-Value
Treatment used *, *n* (%)					
Herbs	719 (98.2)	191 (97.0)	205 (98.1)	323 (99.0)	0.199
Shea butter	34 (4.6)	14 (7.1)	1 (0.5)	19 (5.8)	0.0025
Maraboutage Other practices	27 (3.7) 11 (1.5)	9 (4.6) 4 (2.0)	6 (2.9) 3 (1.4)	12 (3.7) 4 (1.2)	0.670 0.775

* Results refer to the number of times a treatment was cited as one of the two most frequent.

**Table 4 ijerph-16-04587-t004:** Reasons for collaboration between the traditional healers and modern medicine.

Descriptive Variables	Total *n* = 231	Kayes *n* = 87	Koulikoro *n* = 56	Sikasso *n* = 88	*p*-Value
Clinical diagnosis, *n* (%)	68 (29.4)	39 (44.8)	17 (30.4)	12 (13.6)	<0.001
Para-clinical diagnosis, *n* (%)	11 (4.8)	1 (1.1)	7 (12.5)	3 (3.4)	0.006
Treatment, *n* (%)	209 (90.5)	85 (97.7)	42 (75.0)	82 (93.2)	<0.001
Follow-up, *n* (%)	43 (18.6)	6 (6.9)	12 (21.4)	25 (28.4)	0.001

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
