# Peer review of "Sociodemographic Characteristics of Traditional Healers and Their Knowledge of Noma: A Descriptive Survey in Three Regions of Mali"

_ijerph, 2019, doi:10.3390/ijerph16224587_

Round 1

Reviewer 1 Report

In this ms, the authors present an interesting series to study the sociodemographic characteristics of traditional healers to evaluate the knowledge of noma in three regions in Mali as well as the approximate reported number of noma cases. The authors address here with an elegant set of analyses which allows them to gain some important insight into the frequency and knowledge of noma in Mali. However, a few details would be helpful to discuss a bit more. Comment on the correlation between the education level of traditional healers and the potential interest to collaborate with modern medicine. And what are the possible risks associated with high noma in the Koulikoro region.

Author Response

Comment 1 :       In this ms, the authors present an interesting series to study the sociodemographic characteristics of traditional healers to evaluate the knowledge of noma in three regions in Mali as well as the approximate reported number of noma cases. The authors address here with an elegant set of analyses which allows them to gain some important insight into the frequency and knowledge of noma in Malil”.

Answer 1 :            Thank you for your appreciation of our study and for the constructive comments provided.

Comment 2 :       “However, a few details would be helpful to discuss a bit more. Comment on the correlation between the education level of traditional healers and the potential interest to collaborate with modern medicine.

Answer 2 :            We believe that lines 216-218 (“These findings allow to hypothesize that future healers could be fewer in number, but with a higher education level and therefore more prone to collaborate with modern medicine.”) already comment on this point. In addition, lines 192-194, “Regarding a potential interest in receiving some education specifically on noma and gingivitis, all (100%) were in agreement, including a willingness to have some collaboration with modern medicine in the case of a suspicion of noma among their patients” indicate that all healers participating in the study were willing to have some collaboration and thus irrespective of their education level.

Comment 3 And what are the possible risks associated with high noma in the Koulikoro region.

Answer 3 :            A sentence has been added to clarify this point (lines 249-251): “In our opinion, the higher prevalence of noma in Koulikoro compared to Kayes and Sikasso, is only apparent and is probably linked to a better knowledge and therefore a better recognition of the disease.”

Reviewer 2 Report

The manuscript titled ‘Sociodemographic characteristics of traditional 3 healers and their knowledge of noma: a descriptive 4 survey in three regions of Mali’ is a descriptive study describing the sociodemographic characteristics of traditional healers in Mali and their knowledge of noma.

The paper is well written and the topic of research is good and very interesting.

With the knowledge that there is a high prevalence of noma in Mali and that poorer rural areas in Mali do not have access to healthcare facilities, one would assume that traditional healers know a great deal on the clinical features and diagnosis of noma.

This study brought to ‘light’ that only 10.5% of traditional healers had some knowledge of noma and that majority of the traditional healers were older therefore showing a decline in the number of new traditional healers.

The study also shed light on reasons why the prevalence of noma is high in Mali is because of the declining number of traditional healers who anyway lack the knowledge of noma and the lack of medical/healthcare facilities in the areas. This further shows the importance of making available trained professionals who have the expertise to prevent, diagnose the disease in its early stages and prevent the development of noma.

The manuscript is well written and can be accepted in its present form.

Minor suggestion:

The authors can discuss the pathogenesis with regard to the progression of ANG to ANP to ANS and then noma.  

Author Response

Comment 1:        « The manuscript titled ‘Sociodemographic characteristics of traditional 3 healers and their knowledge of noma: a descriptive 4 survey in three regions of Mali’ is a descriptive study describing the sociodemographic characteristics of traditional healers in Mali and their knowledge of noma.

The paper is well written and the topic of research is good and very interesting.

With the knowledge that there is a high prevalence of noma in Mali and that poorer rural areas in Mali do not have access to healthcare facilities, one would assume that traditional healers know a great deal on the clinical features and diagnosis of noma.

This study brought to ‘light’ that only 10.5% of traditional healers had some knowledge of noma and that majority of the traditional healers were older therefore showing a decline in the number of new traditional healers.

The study also shed light on reasons why the prevalence of noma is high in Mali is because of the declining number of traditional healers who anyway lack the knowledge of noma and the lack of medical/healthcare facilities in the areas. This further shows the importance of making available trained professionals who have the expertise to prevent, diagnose the disease in its early stages and prevent the development of noma.

The manuscript is well written and can be accepted in its present forminformative

Answer 1 :            Thank you very much for your valuable comments and interest in our study.

Comment 2 :        “The authors can discuss the pathogenesis with regard to the progression of ANG to ANP to ANS and then noma. »

Answer 2 :  A sentence has been added (lines 57-59): “The pathogenesis of the progression from ANG to noma remains unclear. Some bacteria have been described as trigger organisms, but these have not been further confirmed. [5,13]”

Reviewer 3 Report

Overall this is a good paper. Below I have some comments about clarifications that could improve the manuscript.

Introduction: This is a great introduction. I think it would be improved by including more information about traditional healers in general (e.g. demographics from other studies and use of traditional healers) since this is the focus of the paper.

Methods

Data collection:

How was the questionnaire designed? Was it based off an existing one? If not, what steps were taken to ensure validity?

Consider including a few sample items from the questionnaire measuring knowledge.

Data analysis:

Describe the operationalization of all variables. Was knowledge a continuous variable? How was the questionnaire scored?

In the results (line 159-160) types of traditional practices and secondary activities are mentioned. Provide the operationalization of these variables in the methods section.

Results

Figure 3: Was the number of reported cases of noma derived from description of symptoms? How did they providers report seeing noma if there was only about 10% with knowledge of it?

Discussion

In the discussion of limitations mention the survey. Describe social desirability limitations and issues with validity if the questionnaire was developed just for this study.

Author Response

Comment 1:        « Overall this is a good paper. Below I have some comments about clarifications that could improve the manuscript »

Answer 1 :            Thank you very much for your appreciation of our work and constructive comments provided.

Comment 2:        «Introduction: This is a great introduction. I think it would be improved by including more information about traditional healers in general (e.g. demographics from other studies and use of traditional healers) since this is the focus of the paper»

Answer 2 :            Thank you very much for your comment. Some information has now been added as requested (lines 85-91):

“Traditional medicine is the oldest known healthcare system and was the only answer to health problems in Africa before colonization. It is still used by millions of people around the world. Traditional medicine tends to consider a disease as the rupture of an equilibrium, i.e., a harmony between body and soul.[17] The preferential use of traditional medicine can be explained by the lack of any other choice (difficult access to modern health care for geographic or financial reasons) and by cultural beliefs.[18] As shown by Abdullahi, the ratio of traditional healers to the population in Africa is 1:500 compared to 1:40,000 for “modern healthcare workers”.[17] “

Comment 3:        How was the questionnaire designed? Was it based off an existing one? If not, what steps were taken to ensure validity?

Answer 3 : The questionnaire is not based on a previous one and was developed based on our research questions. As mentioned, studies on traditional healers are quite rare and to evaluate our questionnaire we verified it with different local (Malian) health workers who know noma and are used to work with traditional practitioners in order to ensure that our questions would be understandable for our study population.   

Comment 4:        “Consider including a few sample items from the questionnaire measuring knowledge”.

Answer 4 :            Thank you for this comment. However, we do not believe that this would be of any additional interest to the reader as the results provide many details. Please also see our response to comment 5 below.

Comment 5:        “Describe the operationalization of all variables. Was knowledge a continuous variable? How was the questionnaire scored?”.

Answer 5:       Knowledge was a declarative dichotomous variable where the healer answered by “yes” or “no” to questions such as “ I know noma disease”, “I know the different types of ginigivitis”, etc.Variables were described in their collected format without any operationalization. If the editor considers it to be necessary, we are certainly willing to add the questionnaire in an appendix.

Comment 6:        In the results (line 159-160) types of traditional practices and secondary activities are mentioned. Provide the operationalization of these variables in the methods section.

Answer 6:             Respondents had to select all their activities in a list of 7 plus one open answer for “other specialty”. Regarding a secondary professional activity, this was an open question and we reported the most frequently declared secondary professional activity.

Comment 7: Was the number of reported cases of noma derived from description of symptoms? How did they providers report seeing noma if there was only about 10% with knowledge of it?

 Answer 7 : Healers were asked how many ANG they were seeing every month. The answer was a categorical variable in 3 categories (<10, 10-20 and >20 for ANG). For noma, they needed to give the number of noma cases seen per month for acute noma and noma with sequelae. We created a secondary 3 categories of noma cases, 1-5, 6-10 and >10 based on their answers.

Comment 8: In the discussion of limitations mention the survey. Describe social desirability limitations and issues with validity if the questionnaire was developed just for this study.

 Answer 8: Thank you for this comment. We have now added a sentence mentioning the bias of social desirability (lines 274-275): “First, of all, the social desirability bias inherent to any survey design and the inclusion criteria……”